# What is the contribution of physician associates in hospital care in England? A mixed methods, multiple case study

Vari M Drennan,[1] Mary Halter,[1] Carly Wheeler,[1] Laura Nice,[2] Sally Brearley,[3] James Ennis,[2] Jonathan Gabe,[4] Heather Gage,[5] Ros Levenson,[6] Simon de Lusignan,[7] Phil Begg,[8] James Parle[2]

For numbered affiliations see end of article.

**Correspondence to**
Dr Vari M Drennan;
v.drennan@sgul.kingston.ac.uk

## ABSTRACT

**Objectives** To investigate the deployment of physician associates (PAs); the factors supporting and inhibiting their employment and their contribution and impact on patients' experience and outcomes and the organisation of services.
**Design** Mixed methods within a case study design, using interviews, observations, work diaries and documentary analysis.
**Setting** Six acute care hospitals in three regions of England in 2016–2017.
**Participants** 43 PAs, 77 other health professionals, 28 managers, 28 patients and relatives.
**Results** A key influencing factor supporting the employment of PAs in all settings was a shortage of doctors. PAs were found to be acceptable, appropriate and safe members of the medical/surgical teams by the majority of doctors, managers and nurses. They were mainly deployed to undertake inpatient ward work in the medical/surgical team during core weekday hours. They were reported to positively contribute to: continuity within their medical/surgical team, patient experience and flow, inducting new junior doctors, supporting the medical/surgical teams' workload, which released doctors for more complex patients and their training. The lack of regulation and attendant lack of authority to prescribe was seen as a problem in many but not all specialties. The contribution of PAs to productivity and patient outcomes was not quantifiable separately from other members of the team and wider service organisation. Patients and relatives described PAs positively but most did not understand who and what a PA was, often mistaking them for doctors.
**Conclusions** This study offers new insights concerning the deployment and contribution of PAs in medical and surgical specialties in English hospitals. PAs provided a flexible addition to the secondary care workforce without drawing from existing professions. Their utility in the hospital setting is unlikely to be completely realised without the appropriate level of regulation and authority to prescribe medicines and order ionising radiation within their scope of practice.

## BACKGROUND

Healthcare systems internationally are faced with shortages of doctors and constraints on financial resources, set within a context of an ageing and growing global population.[1 2]

## Strengths and limitations of this study

► This is the first study of the contribution of physician associates (PAs) across multiple secondary care specialties in the National Health Service in England.
► A strength was the diversity within and across the six case study hospitals, including size, socioeconomic setting, secondary and tertiary care specialties and geographical location in three regions in England.
► The mix of qualitative and quantitative methods gathered and synthesised data from multiple perspectives and sources, supporting the trustworthiness and credibility of the findings.
► The difficulty of attributing processes, outcomes and costs to the inclusion of one specific professional in team-based acute clinical care limited our analysis in part.

The combination of these factors has resulted in many countries developing mid-level, or advanced clinical practitioners (ACP).[3] ACPs have often been developed from the nursing workforce but in many countries there are other types of ACP roles; one such group are physician assistants known as physician associates (PAs) in the UK.[4]

PAs originated in the USA in the 1960s and have spread to other countries such as Canada and the Netherlands.[4 5] PAs are a new and rapidly growing occupational group in the UK National Health Service (NHS).[6] PAs are trained at a postgraduate level using the medical model to work in all settings and undertake medical histories, physical examinations, investigations, diagnosis and treatment within their scope of practice as agreed with their supervising doctor.[7] Currently, UK PAs cannot prescribe medicines or order ionising radiation, unlike PAs in countries such as the USA and the Netherlands.[5 8] The Department of Health , following a public consultatiion in 2018, has agreed to regulate PAs.[9] PAs working in primary care in England have been found to complement the work of

general practitioners and to be acceptable, appropriate, safe and efficient.[10] Patients reported PAs to provide good quality care but they did not all understand the role.[11] About 75% of PAs in the UK work in secondary care[12]; however, little is known about their contribution or impact. Pilot projects with American trained PAs working in the UK in 2006 and 2008 concluded that PAs assisted medical teams safely, worked at clinical assistant level and were well received by patients.[13 14] By 2015, 30 of 201 English NHS hospital trusts (the English term for an NHS provider organisation) were employing PAs[15] and a survey of medical directors in 2016 reported that 44 of 71 respondents were considering employing PAs.[16] While the spread of PAs in English hospitals suggests the role is seen as advantageous, there was little evidence available as to the deployment, acceptability, effectiveness and costs of PAs. This paper reports on an investigation into the deployment, acceptability and impact of PAs in a purposive sample of six acute care hospital organisations in England. This investigation was part of a larger multiphase study.[17] The research questions addressed in the investigation were: how are PAs deployed in hospital medical and surgical teams and what supports or inhibits their inclusion? What is the contribution and impact of including PAs in hospital medical and surgical teams on the patients' experience and outcomes, on the organisation of services, working practices and relationships between professionals?

## METHODS

A mixed methods approach was undertaken in 2016–2017, using a case study design[18] in a purposive sample of six NHS acute care hospital trusts in England which employed PAs. The theoretical framing for the study drew on the work of Donabedian in assessing quality in healthcare using the dimensions of effectiveness, appropriateness, efficiency, acceptability and safety.[19] The study is reported using the consensus standards for organisational case studies (online supplementary file 1).[20]

Potential case study sites were identified through a national survey of medical directors who indicated initial willingness to participate.[16] Final decisions were based on: achieving diversity in geographical location, size and type of acute hospitals, the willingness of PAs, consultants and managers to volunteer to participate and, in order to ensure anonymity of individual participants, the same medical or surgical specialties had volunteer PAs in at least two case study sites. Chief executives and/or medical directors gave permission for the organisation to participate in the research. Characteristics of the case study sites in three regions of England are provided in table 1 (adapted from Drennan et al[17]).

Invitations for individual PAs to volunteer participate were through a combination of email from the organisations' lead clinicians for PAs and an onsite meeting for PAs and their consultants, called by the medical director or another lead clinician. At this meeting the research team presented the study, answered questions and invited potential volunteers to provide contact details to which more information could be sent, including consent to participate forms.

## Data collection

Data collection comprised semi-structured interviews, PA self-report work logs, observations of PAs, review of organisational documents and requests for routine management information (data, reports, audits) on the work or impact of the PAs.

Semi-structured interviews were conducted with executive level managers, lead consultants, PAs and members of the healthcare teams in which PAs worked (medical, nursing, other staff) and patients and/or their relatives. Information and invitations to participate were sent to executive level managers by using publicly available contact details or via the medical director or named lead clinician for PAs at each site. Consultants and PAs approached other staff members in the first instance for permission for the research team to invite them to participate or the research team provided information and invitation directly through meeting staff members while conducting observations. All patients and relatives were approached in the first instance by the clinical team

| Table 1 | Case study site characteristics | | | |
|---------|-------------------|----------------|-------------------|------------------------------------------------------|
| Hospital | Inpatient beds | Average full time equivalent doctors* | Annual income† | Type of location (Office of National Statistics classification)‡ |
| 1 | 1000+ | >1001 | >£500 million | Urban with major conurbation |
| 2 | 601–800 | <250 | <£200 million | Urban with city and town |
| 3 | 601–800 | 501–1000 | £201–500 million | Urban with city and town |
| 4 | 1000+ | >1001 | >£500 million | Urban with significant rural |
| 5 | 601–800 | 251–500 | £201–500 million | Urban with major conurbation |
| 6 | 201–400 | 251–500 | £201–500 million | Urban with major conurbation |

*Source: NHS Digital, NHS Hospital & Community Health Service (HCHS).[35]
†Publicly available hospital annual reports.
‡Source: Department for Environment, Food & Rural Affairs.[36]

to request permission for the research team to provide information and invitation.

Senior clinicians and managers were asked questions on factors inhibiting and supporting employment of PAs, impact on the service organisation and patient outcomes as well as costs. Professionals were asked about deployment of PAs, acceptability, impact on working practices and contribution to patient experience. The topic guides for interviews of patients and relatives included questions about their experience as well as about the acceptability of the role. The topic guides are given as online supplementary file 2. With permission, interviews were digitally recorded, or notes taken if preferred. Recordings were transcribed and anonymised. Thematic analysis was conducted using a constant comparative method by research team members and patient representatives.[21] First, a sample of transcripts of different types of participants were read and open coded by five members of the team. In discussion the open codes were then grouped into axial codes; both levels then formed the first draft coding framework. This framework was then discussed by the whole research team using a second sample of transcripts and the coding framework adjusted. All transcripts were then analysed through the final coding framework using the NVIVO V.11 software (QRS International).

PAs were invited to complete a 7-day work log up to three times over the period of the study. These work logs were adapted from a previous study of PAs in England.[22] Data of the PA activities, the setting for the activity and time spent on each during each shift recorded were entered into Microsoft Excel and analysed descriptively.

PAs were invited to be observed for up to three sessions by a researcher. For any PA volunteering, permission was also sought from their supervising consultant. The observations drew on the ethnographic tradition.[23] PAs sought assent from patients for the researcher to be present. Field notes of the PAs' activities and interactions were later written up and analysed.[24]

Documentary analysis was undertaken of publicly available annual reports, board minutes and strategies.[21] Participant managers and clinicians were invited to share any relevant internal data or reports that could assist in answering the research questions, for example, patient throughput and outcome data and expenditure on medical locums. The intention, if data were available, was to compare before and after PAs were employed in a particular service.

### Patient and public involvement

The research questions and study design were initially informed by patient and public input from a previous study of PAs in primary care.[10 11] Patient and public involvement (PPI) in this study included: a PPI representative (SB) was a co-applicant and member of the research team, the study advisory group had two PPI representatives and was chaired by SB, PPI forums were established whose members gave input into research materials, interpretation of findings and dissemination.

| Table 2 | Participants interviewed |
| --- | --- |
| **Type of participant** | **Number interviewed** |
| Executive level managers and clinicians | 18 |
| Physician associates (PAs) | 41 |
| Patients and relatives | 28 |
| Consultants (including those with lead responsibilities for PAs) | 24 |
| Junior doctors | 17 |
| Operational managers | 11 |
| Nurses | 28 |
| Other types of staff, eg, allied health professions | 8 |
| Total | 175 |

All PPI representatives attended an emerging findings seminar and received summaries of the findings.

## FINDINGS

The six case study sites employed approximately 70 PAs, and were recruiting more, in a wide range of adult and paediatric specialties. Forty-three PAs participated in the study, working in 13 adult and paediatric specialties (including emergency departments). PAs provided data through interviews (n=41), observations (n=82 sessions of 35 PAs) and work logs (18 PAs provided 107 days). The PAs had been qualified between 1 and 9 years. A total of 175 interviews were conducted across the sites (table 2).

Annual reports, workforce strategies and board minutes were collected for each hospital for the period of the study (n=139). The managers and clinicians were unable to provide any internal service or patient level data to assist in answering the research questions.

Findings from the different types of data have been combined to address the three research questions: the factors supporting and inhibiting their employment; the deployment of the PAs and their contribution and impact on patients' experience, outcomes and the organisation of services.

### Factors influencing the decisions to employ PAs

The evidence here is drawn from the executive level interviews, from senior clinicians and operational managers and from documentary analysis. Necessity was the most commonly cited reason by the executive and senior clinicians for hospitals beginning to employ PAs in order to address four problems:

1. A decrease in the availability of junior doctors with a consequent over-reliance on locum doctors to cover medical shifts, with attendant concerns about high costs and patient safety.
2. Junior doctors not being able to undertake their training activities as they were being diverted to cover ser-

vice rota gaps—an issue that had been the subject of a deanery review in one hospital.

3. An increase in patient workload and consequent challenges in ensuring sufficient doctors available to cover the inpatient wards.

4. A need to improve the quality of care, this included the necessity to improve the quality performance of hospital as assessed by the Care Quality Commission.

The following exemplar illustrates the multiple factors leading to decisions to employ PAs:

> It was a very acute experience for us here at (name of hospital)…it began with a significant reduction in the number of junior doctors that we had available for our rotas, and we were getting increasingly concerned with lack of deanery appointments being made, last minute vacancies arising and a heavy reliance on locums. Also, at the same time we were (name of external assessment which reported problematic quality) with a lot of significant scrutiny and I recall my consultant team being very concerned about the whole integrity of the rota and continuity of medical workforce. ID23 chief executive

The employment of PAs was only one among multiple workforce strategies being employed in all the hospitals to address these problems. Using PAs was considered advantageous as it did not deplete other staff groups.

> The workforce in every department is limited and so to move pharmacists or nursing staff across to do what have traditional previously been medical roles, means that we're then robbing another profession of their workforce which they desperately need as well …and that's why we've looked to recruit PAs. ID46 medical director

In most of the hospitals, the impetus to employ PAs had come from individual specialties within clinical directorates, rather than an executive led initiative. At the start of the study period, two of the hospitals had documented executive level support and engagement with the introduction and education of PAs as well as production of public information about their PAs. By the end of the study, five of the hospitals had documented executive level support for workforce planning strategies, which included increased PA numbers in support of their medical establishment.

The chief inhibiting factor to PA employment stated was the lack of regulation and attendant lack of authority to prescribe medicines and order ionising radiation.

> Prescribing is the Achilles' heel of the physician associate; not being able to prescribe has meant that their essential contribution of hours has been less than we would have wanted it. ID48 medical director

However, for many, this lack of regulation was reported as an issue to be addressed rather than an absolute inhibitory factor:

> The only, the one challenge we have of course is the prescribing issue, or the lack of prescribing, yes, but no, generally, they're (the PAs) a very helpful, positive addition to our staffing. ID55 chief operations officer

> Now quite frankly it is absolutely bonkers to me that they (PAs) can put, you know, a chest drain in a patient but they can't prescribe paracetamol…we need to do this (regulation) for this group of people (PAs) and just get on with it. ID28 chief executive

While negative attitudes of some senior doctors and nurses were reported as initial inhibiting factors to employing PAs, this was reported to change over time as PAs became part of teams and demonstrated what they could contribute. Many senior staff reported that interest in having PAs in medical/surgical teams spread among the consultants once they observed PAs in other teams and at work.

### The deployment of PAs

The evidence here is drawn from the work logs, interviews and observations. The PAs described themselves as belonging to the medical/surgical team, and their place in work rotas reflected this, with their main working hours being daytime on weekdays. Most PAs described their main work taking place on the ward or unit and this was evident from the work logs and observations. Only a small number of PAs undertook any work in outpatients or operating theatres and if so for a small percentage of their time (figure 1).

The core role of the PAs in all adult and paediatric specialties (apart from those working in the emergency department) was to undertake ward-based work for the medical/surgical team (table 3). This ward-based work was described and observed to include: participating in and following up ward rounds and patient reviews led by doctors; clerking and assessment of patients; preparing for, responding to requests and concerns about patients from nursing staff and communicating with patients and relatives. Twenty and 18 per cent of the PAs time, working in surgical and medical specialties (adult and paediatric excluding those in the emergency department),

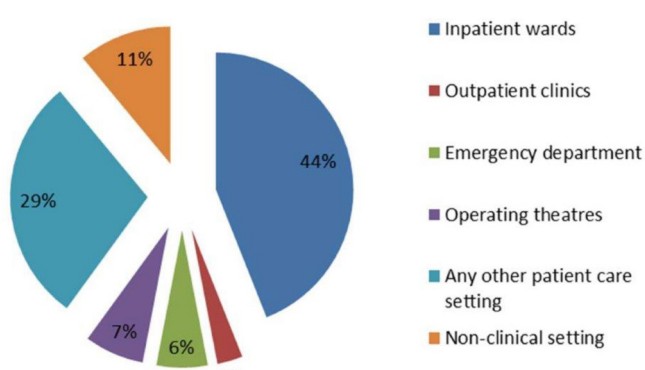

**Figure 1** Working setting for physician associates (PAs) as a percentage of their work hours recorded on worklogs.

**Table 3** Physician associates' time (recorded in work logs) spent on individual work activities, by surgical and medical specialties

| Activity | Surgical specialties* | | | Medical specialties† | | |
| | Time spent on activity | | | | | |
| | Total hours | Mean (SD) weekly hours | Percentage of total hours overall‡, % | Total hours | Mean (SD) weekly hours | Percentage of total hours overall‡, % |
|---|---|---|---|---|---|---|
| Inpatient ward round (with consultant/registrar) | 84 | 7.64 (4.07) | 14 | 61 | 6.1 (2.95) | 15 |
| Inpatient ward round (independent) | 13.25 | 1.2 (1.59) | 2 | 57.75 | 5.78 (3.07) | 14 |
| Inpatient clerking of new patients | 30.25 | 2.75 (2.82) | 5 | 19.5 | 1.95 (1.79) | 5 |
| Inpatient reviewing patients | 42.5 | 3.86 (4.8) | 7 | 69.75 | 6.98 (6.23) | 17 |
| Inpatient preoperative/postoperative assessment | 14.5 | 1.32 (1.8) | 2 | 0.5 | 0.05 (0.16) | <1 |
| Inpatient discussion of patient care/case management with clinical colleagues | 41 | 3.73 (2.84) | 7 | 27.25 | 2.72 (3.18) | 7 |
| Outpatient clerking new patients | 3 | 0.27 (0.9) | 1 | 1.5 | 0.15 (0.34) | <1 |
| Outpatient patient consultation | 23.25 | 2.11 (3.02) | 4 | 3.75 | 0.38 (1.19) | 1 |
| Outpatient preoperative assessment | 0.75 | 0.07 (0.23) | <1 | – | – | – |
| Outpatient discussion of patient care/case management with clinical colleagues | 0.5 | 0.05 (0.15) | <1 | 4.25 | 0.42 (1.34) | 1 |
| Emergency department clerking new patients | 3.25 | 0.3 (0.43) | 1 | 5 | 0.5 (1.58) | 1 |
| Emergency department patient consultation | 1.25 | 0.11 (0.26) | <1 | – | – | – |
| Emergency department discussion of patient care/case management with clinical colleagues | 0.75 | 0.07 (0.16) | <1 | 5 | 0.5 (1.58) | 1 |
| Assisting in theatre/interventional procedures | 66.25 | 6.02 (6.28) | 11 | 11.25 | 1.12 (1.46) | 3 |
| Patient education (any setting) | 14.25 | 1.3 (1.52) | 2 | 8 | 0.8 (1.25) | 2 |
| Discussing care with relatives (any setting) | 12.5 | 1.14 (1.07) | 2 | 28.75 | 2.88 (2.21) | 7 |
| Routine procedures (eg, phlebotomy, cannulation, ECG) (any setting) | 34.25 | 3.11 (2.13) | 6 | 26.75 | 2.68 (1.2) | 7 |
| TTOs and discharge summaries (any setting) | 59 | 5.36 (4.4) | 10 | 34.5 | 3.45 (2.37) | 8 |
| Requesting investigations (any setting) | 33.75 | 3.07 (2.78) | 6 | 20.75 | 2.08 (1.65) | 5 |
| Administration | 24.25 | 2.2 (3.82) | 4 | 2.5 | 0.25 (0.58) | 1 |
| Teaching | 22.5 | 2.04 (2.49) | 4 | 10.5 | 1.05 (1.46) | 3 |
| Own training/study | 14.5 | 1.32 (2.22) | 2 | 2.5 | 0.25 (0.49) | 1 |
| Networking/attending meetings | 15.5 | 1.41 (1.86) | 3 | 1.75 | 0.18 (0.37) | <1 |
| Strategy/policy/service development | 2.75 | 0.25 (0.58) | <1 | 2.5 | 0.25 (0.79) | 1 |
| Other§ | 31.75 | 2.89 (4.24) | 5 | 1 | 0.1 (0.32) | <1 |
| Total | 589.5 | 53.59 (11.68) | n/a | 406 | 40.6 (12.78) | n/a |

*n=10 participants.
†N= 6 participants.
‡Due to rounding, percentages may not add up to 100%.
§'Other' activities included: collecting notes, surgical planning meeting, interpreting investigations, ward list/preoperative, multidisciplinary team, airway support and assisting intubation, telephone clinic, university teaching.
TTO, to take out.

respectively, was spent in ordering tests, preparing discharge summaries and administration (table 3).

In the emergency department setting, the PAs worked in the major, and sometimes minors, sections where they were described and observed to be assigned to undertake patient assessments (following clinical triage) and, as agreed by the senior supervising doctor after presentation of each patient assessment, order investigations and formulate management and treatment plans. They were observed to work as part of the multidisciplinary team alongside junior doctors and other advanced clinical practitioners, who reported similarly to the senior doctor.

Individual PA roles were described and observed to be moulded to the need of a service and that over time some PAs had been trained to undertake procedures common for that specialty such as lumbar punctures, echocardiograms, peripherally inserted central lines or nerve blocks.

Consultants and managers reported that PAs were primarily deployed to help address gaps in medical rotas.

## The contribution and impact of including PAs in medical/surgical teams to patients' experience

Patients and relatives reported very positive views of the PAs attending them. Particular aspects mentioned were: the PA's constant presence on the ward meant they were easy to approach and PAs followed up items from the doctor's ward round and spent time explaining decisions and management plans to the patients and relatives.

> … he's (the PA) been quite instrumental in helping me understand things because when doctors come they say things that people are writing down and then they walk away and you find out that they have changed your medication and obviously I need an answer as to why, so I go to him and he explains. ID143 patient

Many of those interviewed and observed were uncertain about what a PA was or mistook them for a doctor, despite the PA or the consultant introducing them as a PA:

> I thought she was a doctor. But is she? ….she came to see me, and I was perfectly happy with her expertise and everything else, so I don't want to give the wrong impression. But I had other things on my mind (than) to ask what her actual title was. ID106 patient

All of the patients and relatives reported that they saw the PAs working within the medical/surgical team, PAs provided good care and referred back to senior doctors. All patients and relatives interviewed were content to be attended to by a PA in the future.

Many of the health professional and managerial participants voluntarily offered information on the high volume of compliments and presents the PAs received from patients.

## The contribution and impact on outcomes and the organisation of services of including PAs in medical/surgical teams: the professionals' view

The majority of doctors, nurses and managers described the contribution of PAs as positive.

> They work alongside our junior doctors, support the junior doctors, and are clinically very valuable. Well, almost invaluable now. ID185 consultant

> They're (PAs) a really valuable asset in the department now and we're looking at expanding, seeing how we can have more (PAs) in the department to help. ID205 operational manager

A small number of doctors and nurses in high dependency specialties considered that, having employed or worked with PAs, doctors were more suited to the work of the specialty. The extent to which the PAs' lack of authority to prescribe was influential in this was unclear.

The reported positive contribution of PAs was grouped into themes of: providing continuity, aiding patient flow, supporting patient safety and releasing doctor time for more complex patients and training, which we now discuss in turn.

## Continuity

One of the most frequently reported impacts on the organisation was that PAs provided continuity of staffing in the medical/surgical team, that is, personal and team continuity. They provided continuity in presence and continuity in knowledge and relationships which was reported as beneficial to patients, nurses and doctors in these ways:

► Continuity in presence on the inpatient wards which increased access and early escalation of problems to the medical/surgical team for nurses: "*If we need any form of escalation, getting in touch with doctors, we can also get in touch with the PAs, the PAs chase the doctors, so their role is quite significant as well. …to get things going so patients are not left for long hours waiting for a doctor because doctors are doing other things, doctors are in theatres. They're like the middle person who get things done between both sides, nurses and doctors*". ID71 senior nurse

► Continuity in knowledge about current inpatient status, management plans and patients' progress, which facilitated updating patients and the medical/surgical team: "*It's that continuity, like I've been on call so I've not been on the ward for 5 days, I'm like 'what's going on?' and they're (PAs) like 'seven and five are the sick ones and ten is the one we're trying to get out'. And like they know that and that's very helpful*". ID207 foundation year doctor

► Continuity in knowledge about the policies and practices (clinical and otherwise) of the department, the individual consultants and the hospital which was reported to be of particular value for doctors on short training rotations new to that particular workplace. "*Our SHO equivalent doctors rotate all the time… so what they (the PAs) really provide is this amazing continuity of how the system works… how we care for patients who've had* (specialty) *procedures, how we manage patients with different* (specialty) *conditions*". ID51 registrar

## Aiding patient flow

PAs were described and observed to undertake large amounts of non-patient facing clinical work for the medical/surgical team. All participants reported PAs helped smooth and improve patient flow.

> to liaise within teams, to liaise with other departments, to book a test, to get in touch with a GP, to book a bed for a patient which (sic) is frail and elderly, which (sic) need antibiotics a few days before, comes from far away, … and they (the PAs) can really help for that organisational aspect a lot, and also help in the more clinical aspect… I think they (the PAs) smooth things out with many issues that need to be prepared and planned for. ID95 registrar

> PAs provide ward cover so discharge summaries are completed on time, meaning patients leave hospital without delay and bed capacity is released for other patients. ID81 operational manager

One manager described the PAs as 'oil' in the system, while a consultant likened the work of the PAs to 'the glue' within the medical/surgical team.

### Releasing doctors' time

The presence of a PA in the team was considered to release the doctors' time to attend more complex patients and also to attend patients in outpatients and theatre.

> They're (PAs) just great at coming in and just taking off those little jobs that will really slow you down unnecessarily and paving the way for the more important sicker patients to get more of your time and attention. ID12 core training doctor

However, caveats were offered in that in some specialties efficiencies of the role were not fully realised due to lack of authority to prescribe.

> It's (lack of authority to prescribe medicines) a real hindrance, because somebody's then got to do it. So we want to discharge a patient, we've all agreed that's what they need doing, somebody of course needs to write up their drugs… the PA's doing all the discharge planning and everything else, but can't do this bit, so then has to wait for a junior doctor to come along and do it. ID185 consultant

Some registrars gave estimates that the PAs, without authority to prescribe, could cover about 70% of the work required.

### Patient safety

All consultants, registrars and managers reported the PAs to be safe with no serious incidents or patient complaints. Many of the doctors reported that the PAs were careful to work within their capabilities and within guidelines and appropriately refer to the doctors within the team. The continuity PAs provided, as described above, was viewed as important in patient safety. In some services, the PAs' duty times were arranged to cover for absences of doctors, for example, to attend training and reduce the use of locum doctors. Consultants and managers considered locum doctors that were new to their service as less efficient, less safe and costlier than PAs.

> Better for patient safety to have the PAs than using people that you don't know, locums coming in can create chaos. ID114 consultant

### QUANTIFYING THE IMPACT

When asked to quantify the impact of the PAs, all the senior clinicians and managers pointed out that it was hard to separate out the individual from the overall large, multidisciplinary team(s) delivering acute care. Those that were able to offer views did so anecdotally describing reduction in spending on locum doctors, improved use of senior clinicians' time and greater productivity of the medical/surgical team:

> We have had to spend more money on (locum) doctors when we don't have PA cover, just to double up so it's safe. ID203 operational manager

> PAs assisting in theatres—this has seen a reduction in theatre cancellations and increased efficiency due to a lack of junior doctors being available to assist. Resulting in reduced wait times and complaints, and income generation. ID81 operational manager

### DISCUSSION

This study offers new insights as to the deployment and contribution of PAs in a range of medical and surgical specialties in English NHS hospitals. PAs were found to be acceptable, appropriate and safe members of the medical/surgical teams by the majority of the doctors, managers and nurses. They were mainly deployed to undertake inpatient ward work in the medical/surgical team during core weekday hours. They were reported to contribute positively to continuity in the medical/surgical team, to patient experience and flow, to inducting new junior doctors and supporting the medical/surgical teams' workload thus releasing doctors for attending the more complex patients and for their training. The continuity that PAs brought to medical/surgical teams was viewed as more useful and safer than employing locum doctors, who were also reported as costlier to the service than PAs. There were suggestions that some PAs increased senior clinicians' productivity. The finding of PAs practising safely has also been reported in a systematic review.[26] Observations on the continuity that PAs provide within medical/surgical teams have been made in North America and the Netherlands[27 28] and in the USA regarding releasing doctors to undertake training.[29]

Patients and relatives reported that they viewed PAs and their contribution positively but most did not understand what a PA was. Patients have been reported to have higher levels of satisfaction from hospital teams which included PAs compared with those without in the Netherlands.[30] Patient confusion about the PAs role has been reported before in the primary care setting.[11]

The contribution of PAs to productivity, patient experience and outcomes was not quantifiable separately from other members of the team and wider service organisation. This finding has also been reported in the USA.[31] The lack of authority to prescribe and order ionising radiation was reported as an inhibiting factor to their employment and meant that in some specialties the full potential of PAs could not be realised. A Dutch study, where PAs have authority to prescribe, reported no difference in cost-effectiveness of inpatient care between teams with and without PAs.[32]

### Strengths and limitations

The strength of this study was that it was undertaken in different types of hospitals, across different specialities and a large number of PAs. The mixed methods case study approach gave qualitative insights from a wide range of

stakeholders and illuminated the quantitative data from PA work logs. The main limitations were the low response rate in providing work logs, which we understood was a result of workload pressures, and that participants were either not aware or were unable to access and provide to the research team quantitatively measurable data on the effect of the introduction of PA on patient outcomes or human resource expenditure. In addition, attribution of changes in quality of patient care, patient safety events or cost savings to the introduction of PAs was problematic when such introduction commonly went alongside other changes or response to other significant events such as unprecedented surges in demand and workforce shortages.[33][34] However, we were able to describe and scope the range of potential impacts. These might be explored in more depth in future studies using matched comparisons of medical/surgical teams as in a Dutch study,[32] or in a 'step-wedge' design (where the change is introduced sequentially in all sites so that all 'participants' get the intervention, but not simultaneously).

## CONCLUSION

Planning and developing a medical and surgical workforce can be challenging. Achieving a sustainable medical workforce with the right balance of consultants, specialty doctors (ie, not in training posts) and junior doctors has to take account of: (a) the service demands, (b) the training requirements for career advancement and (c) the creation of manageable jobs within interesting careers. Having a cadre of flexible, advanced clinical practitioners trained in the medical model, who can support doctors may help address inherent tensions between service demands, training requirements and budgetary constraints, as well as mitigate against cyclical shortages. PAs could provide a flexible addition to the secondary care workforce without drawing from existing professions. PAs may provide personal and team continuity not provided by junior doctors, who frequently rotate to new training posts. Many experienced clinicians valued this continuity more highly than the service delivered by locum doctors unfamiliar with the setting. However, PAs' utility in the hospital setting is unlikely to be fully realised without the appropriate level of regulation with attendant authority to prescribe medicines and order ionising radiation within their scope of practice.

**Author affiliations**
[1]Centre for Health and Social Care Research, Joint Faculty of Kingston University and St. George's University of London, London, UK
[2]Institute of Clinical Sciences, University of Birmingham, London, UK
[3]Centre for Public Engagement, Joint Faculty of Kingston University and St. George's University of London, London, UK
[4]Department of Criminology and Sociology, School of Law, Royal Holloway, University of London, Egham, UK
[5]Surrey Health Economics Centre, Department of Clinical and Experimental Medicine, University of Surrey, Guildford, UK
[6]Independent consultant, London, UK
[7]Department of Clinical and Experimental Medicine, University of Surrey, Guildford, UK
[8]Royal Orthopaedic Hospital, Birmingham, UK

**Acknowledgements** The authors would like to thank all the patient and staff participants who generously gave their time and views during a period when there were many pressures and demands on NHS services.

**Contributors** Conception, design and receiving funding for the study: VMD, MH, SB, SdL, HG, JG, PB, JE, JP. Lead for PPI: SB. Acquisition of the data: CW, LN, JE, RL, MH, VMD. Analysis and interpretation of the data, drafting and critical revision of the manuscript for important intellectual content, accountable for all aspects of the work and approval of the final manuscript: VMD, MH, CW, LN, RL, SB, SdL, HG, JG, PB, JE, JP.

**Funding** This project was funded by the National Institute for Health Research Health Services and Delivery Research Programme (project number 14/19/26). This paper presents independent research commissioned by the National Institute for Health Research (NIHR).

**Disclaimer** The views and opinions expressed by authors in this publication are those of the authors and do not necessarily reflect those of the NHS, the NIHR, the Health Service and Delivery Research Programme or the Department of Health.

**Competing interests** SdL is head of the Department of Clinical and Experimental Medicine at the University of Surrey, which launched a physician associate course in 2016. JP chairs the UK and Ireland Board for Physician Associate Education and was director of the physician associate course at the University of Birmingham. PB is honorary faculty at the University of Birmingham and has taught on the physician associate programme since 2008. JE teaches part time on the University of Birmingham physician associate course.

**Patient consent for publication** Not required.

**Ethics approval** The NHS Health Research Authority approval (IRAS project ID: 181193) and NHS London Central Research Ethics Committee (REC reference: 15/LO/1339) approval were obtained.

**Provenance and peer review** Not commissioned; externally peer reviewed.

**Data sharing statement** No additional data are available.

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
