## [Reviewer comments · BMJ Open]

This paper was submitted to a another journal from BMJ but declined for publication following peer review. The authors addressed the reviewers' comments and submitted the revised paper to BMJ Open. The paper was subsequently accepted for publication at BMJ Open.

(This paper received three reviews from its previous journal but only two reviewers agreed to published their review.)

ARTICLE DETAILS

TITLE (PROVISIONAL)	WHAT IS THE CONTRIBUTION OF PHYSICIAN ASSOCIATES IN HOSPITAL CARE IN ENGLAND? A MIXED METHODS, MULTIPLE CASE STUDY.
AUTHORS	Drennan, Vari; Halter, Mary; Wheeler, Carly; Nice, Laura; Brearley, Sally; Ennis, James; Gabe, Jonathan; Gage, Heather; Levenson, Ros; de Lusignan, Simon; Begg, Phil; Parle, James

VERSION 1 – REVIEW

REVIEWER	Asa Auta University of Central Lancashire, United Kingdom
REVIEW RETURNED	05-Nov-2018

GENERAL COMMENTS	This article reads well, it highlights the contributions of PAs in hospital care in England and the challenges associated with their roles. The methodology is sound and interpretations/conclusions were focussed on the data obtained. Please find below a few comments to consider. Methods: Page 7 lines 6 – 14. Can the authors please include as supplementary materials, the interview guides used for this study? Page 7 line 19. There are several ways of conducting thematic analysis. Can the authors briefly describe how thematic analysis was conducted in this study? Page 7 line 28. How many PAs were observed? How were participants recruited in this study? Findings Factors influencing the decisions to employ PAs While the evidence from this theme were drawn from a range of stakeholders including senior clinicians and operational managers, only quotes from senior clinicians were used as supporting evidence for interpretations. Can authors please use quotes from a variety of stakeholders to support data interpretations? The contribution and impact on outcomes and the organisation of services of including PAs in medical/surgical teams: the professionals' view Page 13 lines 17 – 20 – Can authors provide a quote from a participant to support the findings reported here?
--

	Releasing doctors time Page 14 lines 53-54. Lack of authority to prescribe was identified as one of the key barriers to the role of PAs. Can authors provide a supporting statement from participants to support the findings here?
--	--

REVIEWER	James F. Cawley, MPH, PA-C, DHL(hin) Professor The George Washington University Washington, D.C. USA
REVIEW RETURNED	19-Nov-2018

GENERAL COMMENTS	General Comments The paper is an important descriptive report on the utilization of UK PAs in the inpatient hospital setting. Its strengths include that its approach consists on both quantitative as well as qualitative methods and included several types of inpatient facilities. While the paper contains findings observed elsewhere related to PA utilization, acceptance, and effectiveness in the inpatient hospital setting, the paper provides additional affirmation of PA contributions in the UK where inpatient utilization of PAs is still in its early stages. Specific Comments  1. Data tables in general are clear and well organized 2. The number (70) of study subjects and hospitals are rather small;this is acknowledged by the authors as limitations. However, the use of multiple mixed methods coupled with a good deal of detail in the information gathering compensates for this shortcoming. 3. Page 16, 2nd paragraph. One finding that is included in the paper that expresses a widely-held view but is not well documented in the US PA health services research literature is the finding that "Patients and relatives described PAs positively but most did not understand who and what a PA was, often mistaking them for doctors."
--

VERSION 1 – AUTHOR RESPONSE

Comment	Response
Reviewer: 1	
This article reads well, it highlights the contributions of PAs in hospital care in England and the challenges associated with their roles. The methodology is sound and interpretations/conclusions were focussed on the data obtained. Please find below a few comments to consider.	Thank you

Comment	Response
Methods:	
Page 7 lines 6 – 14. Can the authors please include as supplementary materials, the interview guides used for this study?	We have inserted in the text The topic guides are given as supplementary file 2. We have added a supplementary file 2.
Page 7 line 19. There are several ways of conducting thematic analysis. Can the authors briefly describe how thematic analysis was conducted in this study?	We have re-arranged and added to this section to read : Thematic analysis was conducted using a constant comparative method by research team members and patient representatives. [23]. First, a sample of transcripts of different types of participants were read and open coded by five members of the team. In discussion the open codes were then grouped into axial codes; both levels then formed the first draft coding framework. This framework was then discussed by the whole research team using a second sample of transcripts and the coding framework adjusted. All transcripts were then analysed through the final coding framework using the NVIVO 11 software (QRS International Pty Ltd).
Page 7 line 28. How many PAs were observed?	We have inserted on page where we give all numbers on participants “observations (n= 82 sessions of 35 PAs)”
How were participants recruited in this study?	We have added on page 6 Invitations for individual PAs to volunteer participate were through a combination of email from the organisations' lead clinicians for PAs and an on-site meeting for PAs and their consultants, called by the medical director or another lead clinician. At this meeting the research team presented the study, answered questions and invited potential volunteers to provide contact details to which more information could be sent, including consent to participate forms. We have also added on page 7 Information and invitations to participate were sent to executive level managers by using publicly available contact details or via the medical director or named lead clinician for PAs at each site. Consultants and PAs approached other staff members in the first instance for permission for the research team to invite them to participate or the research team provided information and invitation directly through meeting staff members while conducting observations. All patients and relatives were approached in the first instance by the clinical team to request permission for the research team to provide information and invitation.

Comment	Response
Findings	
Factors influencing the decisions to employ PAs While the evidence from this theme were drawn from a range of stakeholders including senior clinicians and operational managers, only quotes from senior clinicians were used as supporting evidence for interpretations. Can authors please use quotes from a variety of stakeholders to support data interpretations?	We have added the following quotations on page 10 to illustrate the points and ensure a variety of stakeholders are represented. The following exemplar illustrates the multiple factors leading to decisions to employ PAs: “It was a very acute experience for us here at [name of hospital] ...it began with a significant reduction in the number of junior doctors that we had available for our rotas, and we were getting increasingly concerned about the integrity of the rota with lack of deanery appointments being made, last minute vacancies arising and, you know, a heavy reliance on locums on the rota. Also, at the same time we were [name of external assessment which reported problematic quality] so we were going through a lot of significant scrutiny and I recall my consultant team being very concerned about the whole integrity of the rota and continuity of medical workforce,so that was one of the key things” ID23 chief executive And further down the page However . for many this lack of regulation was reported as an issue to be addressed rather than a complete inhibiting factor: “The only, the one challenge we have of course is the prescribing issue, or the lack of prescribing, yes, but no, generally, they're [PAs] are a very helpful, positive addition to our staffing”. ID 55 chief operations officer “ Now quite frankly it is absolutely bonkers to me that they [PAs] can put, you know, a chest drain in a patient but they can't prescribe paracetamol,we need to do this [regulation] for this group of people [PAs] and just get on with it, “ ID 28 chief executive
The contribution and impact on outcomes and the organisation of services of including PAs in medical/surgical teams: the professionals' view Page 13 lines 17 – 20 – Can authors provide a quote from a participant to support the findings reported here?	Our manuscript for this section (p 13 lines 17- 20) is about continuity and we have given supporting quotations to all bullet points in this section.
Releasing doctors time	
Page 14 lines 53-54. Lack of authority to prescribe was identified as one of the key barriers to the role of PAs. Can authors provide a supporting	We have added this additional quotation “It's [lack of authority to prescribe medicines] a real hindrance, because somebody's then got to do it. So we want to discharge a patient, we've all agreed that's what they need

Comment	Response
statement from participants to support the findings here?	doing, somebody of course needs to write up their drugs..... the PA's doing all the discharge planning and everything else, but can't do this bit, so then has to wait for a junior doctor to come along and do it. ID 185 consultant
Reviewer: 2	
The paper is an important descriptive report on the utilization of UK PAs in the inpatient hospital setting. Its strengths include that its approach consists on both quantitative as well as qualitative methods and included several types of inpatient facilities. While the paper contains findings observed elsewhere related to PA utilization, acceptance, and effectiveness in the inpatient hospital setting, the paper provides additional affirmation of PA contributions in the UK where inpatient utilization of PAs is still in its early stages.	Thank you. There were no amendments required by this reviewer.

VERSION 2 – REVIEW

REVIEWER	Dr Asa Auta University of Central Lancashire, Preston, UK
REVIEW RETURNED	06-Dec-2018
GENERAL COMMENTS	Thank you for addressing all the issues raised in the review.